#### 1

# Turbulence measurements with a tethered balloon

CANUT Guylaine<sup>1</sup>, COUVREUX Fleur<sup>1</sup>, LOTHON Marie<sup>2</sup>, LEGAIN Dominique<sup>1</sup>, PIGUET Bruno<sup>1</sup>, LAMPERT Astrid<sup>3</sup>, and MOULIN Eric<sup>1</sup>

<sup>1</sup>CNRM-GAME (Météo-France and CNRS), Toulouse, France

<sup>2</sup>Laboratoire d'Aerologie, University of Toulouse, CNRS, France

<sup>3</sup>Institute of Flight Guidance, TU Braunschweig, Hermann-Blenk-Str. 27, 38108 Braunschweig, Germany

*Correspondence to:* CANUT Guylaine (guylaine.canut@meteo.fr)

# Abstract.

This study presents the first deployment of a turbulence probe below a tethered balloon in field campaigns. This system allows to measure turbulent temperature fluxes, momen-

tum fluxes as well as turbulent kinetic energy in the lower part of the boundary layer. It is composed of a sonic thermo-anemometer and inertial motion sensor. It has been validated 10 during three campaigns with different convective boundary-layer conditions using turbulent measurements from atmo spheric towers and aircraft.

Keywords. Boundary layer, turbulence measurements, tethered balloon

#### 1 Introduction

The atmospheric boundary layer (ABL) is the lowest part of the atmosphere and hosts turbulent processes responsible for transfer of heat, moisture and momentum between the surface and the free troposphere. The time evolution of the parameters close to the surface is controlled by those turbulent processes. Also the coupling to the surface (either land

- or ocean) strongly depends on the boundary-layer processes. So, a precise understanding of those processes and in particular of the vertical profiles of turbulent fluxes is crucial to our ability to quantitatively describe and model the evolution of the lower part of the atmophere and the corresponding
- <sup>25</sup> energy budget, which are necessery for numerical weather and climate predictions. In recent years, dynamics and lower layers exchange of energy and trace species at the surface / atmosphere interface were studied during national and international programs (SHEBA (Utall and al. (2002)), IHOP

(Weckwerth et al. (2004)), AMMA (Lebel et al. (2007)), COPS (Wulfmeyer et al. (2008))).

Understanding the turbulent processes in the ABL requires the knowledge of the evolution of the profile of the sensible heat flux. However, it remains difficult to measure. The observation of these processes raises specific problems because the phenomena involve fine temporal (a few tenths of a second to a few minutes) and spatial scales (of the order of meters to tens of meters). If rapid sensors are available at the ground for most variables (temperature, humidity, wind), in altitude the high-frequency measurements are limited, and the turbulent instruments are mounted mainly on research aircrafts. Previous studies (Lenschow and Stankov (1986), Saïd et al. (2010)) used instrumented aircraft to measure turbulent heat flux in altitude. This platform does not allow to obtain vertical profiles, but only provides some measurements at discrete vertical levels. Usually a linear interpolation of data is used to obtain a profile and estimate fluxes at surface and at the top of ABL. Another inconvenient of aircraft platform is the cost. Recently, studies with remotely piloted aircraft systems (RPAS) (Martin et al. (2014)) show the capability of these small and light platform to measure turbulent heat fluxes in altitude. Fixed-point measurements on tall towers have provided significant insight into the heat fluxes characteristics well above the surface layer (Kaimal et al. (1976); Angevine et al. (1998)) but towers are limited in height with only a few towers worldwide reaching more than 100m. Towers with heights exceeding 50 m are practically non-portable, which makes them inappropriate for deployment in a field campaign. The logistical limitations of other platforms can partly be overcome by using tethered balloons. This platform offers the potential of a fixed mast but can be used to heights up to 1000m and is easily deployed

2

from various locations. Past studies have used this platform since the 1970s. (Morris et al. (1975); Kaimal et al. (1976);

- <sup>10</sup> Ogawa and Ohara (1982); Muschinski et al. (2001)) but this platform has mainly be used to study mean thermodynamical measurements. Lapworth and Mason (1988) developed a system with a turbulence probe composed with a Gill propeller anemometer attached to the tethering cable of a balloon. The
- authors used inclinometers and magnetometers to determine the probe sensor orientation. The system weighted around 10kg.

A detailed examination of the general applicability of an instrumented balloon for measuring ABL turbulent fluxes has

- not been undertaken previously. The objective of this study is to demonstrate that an instrumented balloon can be used for measurements of the heat flux and turbulence structure of the ABL. The major advantage of tethered balloon is the potential to provide flux measurements at various vertical heights
- <sup>25</sup> covering a part of the vertical extent of the boundary layer. The turbulence tethered sonde presented here is designed to measure turbulence and sensible heat fluxes. The paper is structured as follows. First we describe the general architecture of the system, the sensor characteristics and the motion correction. Sections 3 and 4 are dedicated to the validation re-<sup>15</sup>
- spectively close to the surface and within the boundary layer using conventional data from towers and aircraft. In section 5, we explore the capability of the system to study the turbulence structure in the framework of the late afternoon transition. Conclusion ends the paper.

#### 10 2 Overwiew of the system

This part describes the general architecture of the system, i.e. the balloon used and the turbulence sonde. The motivation is to develop a simple device that can be easily deployed in different field campaigns. The platform combines slow and fast sensors to quantify mean and turbulent processes.

#### 2.1 Sensor characteristics

15

In this study we have used the Vaisala 7  $m^3$  tethered balloon inflated with helium. The model is Vaisala TTB327 (L  $_{15}$ 4.6 m x H 1.84 m x l 1.84 m 3.1 kg). The balloon is a zeppelin shaped aerostat and it is restrained by a cable attached to the ground (the weight of the cable is  $0.5 \ 10^{-3} kgm^{-1}$ )

- <sup>5</sup> with an electric winch which is used to raise and lower the balloon. The maximum height of flights that can be reached depends on atmospheric conditions (wind speed). We have never tested an altitude higher 1000 m. The turbulence tethered sonde (denoted TS in the following) can be attached to a
- wide variety of balloon; a dedicated balloon is not necessary. The instrument package consists of both a slow measurement instrument, a 1Hz vaisala tethered sonde (TTS111 model) mounted below the tethered line as well as fast measurement instrument, called the TS and suspended 8 m below the bal-

Figure 1. Image of the turbulence tethered sonde: (a) The sonic anemometer and the electronic system; (b) the inertial motion sensor.

loon to avoid wind flow distortion due to the balloon. The TS is attached to the cable with an horizontal pivot. The advantage is to limit yaw movements of the TS. The 1Hz vaisala commercial probe provides slow measurements of temperature, humidity, pressure, wind speed and direction, and is able to transmit 1Hz data to the ground using a radio link. This probe is mainly used to monitor the wind in real-time at flight altitude. We have a security constraint given by the balloon manufacturer, in case of wind greater than  $12 ms^{-1}$ , when the flight should be interrupted.

The TS is based on a commercial sonic anemometer (Gill windmasterpro model, fig. 1(a)) which provides measurements of three-dimensional wind and sonic-temperature at 10 Hz. The thermo-anemometer allows to connect others sensors to own analog inputs. An off-the-shelf coupled inertial-GPS motion and attitude sensor (Mti-G at 10 Hz from Xsens, fig. 1(b)) was added in order to correct the anemometer movements. A fast-response thin wire allows the measurement of air temperature fluctuations. Also a standard pressure and temperature sensors provide slow reference measurements. Data was logged aboard on two SD cards. A home made data acquisition system (micro controleur PIC24F) read, date, and log thermo-anemometer and inertial navigation system (INS) incoming numerical RS232 signals. The total mass of the system is 2 kg including batteries (0.3 kg). The sonic anemometer represents the half of the mass (1 kq)whereas the GPS-INS weighted only 0.15 kg. A first performance lies in the low weight of the system. Lapworth and Mason (1988) described a balloon borne turbulence probe system with a weight of 10 kg. The decrease in weight was possible by the miniaturization of sensors in recent years. The system can run for 4h powered by eigth 1.2V 2700 mA.h NiMH batteries.

15

3

#### 2.2 Motion correction

The off-the-shelf coupled inertial-GPS motion and attitude sensor is essentially composed of two parts: (1) an inertial navigation system to measure the balloon's position, speed,  $_{20}$  and attitude relative to the Earth, and (2) a data acquisition

system to record all the incoming signals.

A miniature GPS-INS is attached to the platform 40 cm above the sonic anemometer to provide the position, speed, and orientation of the sonic anemometer.

Linear and rotational speeds provided by the INS are used to calculate the speed of the platform in the coordinate system of the sonic anemometer. This means that the wind vector in the platform coordinate system is a simple vector difference between the sonic and GPS-INS velocities:

$$V_{platform} = V_{sonic} - V_{INS} \tag{1}$$

<sup>5</sup> where  $V_{platform}$  is the wind vector in the platform coordinate system,  $V_{INS}$  is the GPS-INS motion vector, and  $V_{sonic}$  is the platform-relative flow vector measured by the sonic anemometer.

The INS measured angles of attitude (rolls, pitch and yaw

angles) allow us rotate the wind vector measured in the platform coordinate system to the meteorological coordinate system. Geo-referenced u, v, w wind components are then calculated from the well adopted equations of Lenschow (1986).

#### 3 Validation close to the surface

- In order to check the validity of the high-frequency measurements obtained by the TS, the measurements are compared with those of a three-dimensional sonic anemometer fixed on masts and installed during three experimental campaigns between 2010 and 2013. Ideally, for direct compari- 15
- son with fixed point on tower, flying at constant altitude close to the tower is desirable. The horizontal distance between TS and the position of the towers was lower than 200 m. The two first campaigns took place in summer 2010 and 2011 in the BLLAST (Lothon et al. (2014)) experimental site with <sup>20</sup> a tower equipped with three-dimensional sonic anemometers (CSAT, Campbell Scientific Inc, Logan, UT, USA) at 60 m and the third took place at Bourges (France) in a french military site which was equipped with a tower with three-
- <sup>5</sup> dimensional sonic anemometers (GILL HS 3-axis, Gill In- <sup>25</sup> struments Limited, Lymington, Hampshire, UK) at 30 m. For all the days considered here, the atmospheric conditions were convective and clear sky. Only the campaign in August 2010 in the BLLAST site was entirely dedicated to the validation
- of the TS. No scientific constraints were therefore imposed. Indeed, during two days, the TS flew at fixed height corresponding to the instrumented level of the mast. For the other two campaigns, the TS did not remain the whole day at the same height. So we only selected measurement periods when
- the TS was at a similar level as the fixed sonic anemometer.

Globally, the time series recorded during these different campaigns, after motion correction applied, exhibit excellent agreement even with the aforementioned spatial differences between tower and TS. We hereafter denote u', v', w' and  $\theta'$ the fluctuations in longitudinal wind, transverse wind, vertical wind and potential temperature respectively. Fluctuations x' of a variable x are computed as:  $x' = x - \bar{x}$  where  $\bar{x}$  is the mean over a chosen period. An example of the high-frequency measurements of fluctuations of the threedimensional winds, and potential temperature is shown in figure 2(a) for a thirty-minute sample on 31 august 2010. The two records do not overlap perfectly but this is expected with fast measurements made 200 meters apart. However, the range of the fluctuations of u, v, w and  $\theta$  are similar between the TS and the data from the fixed sonic anemometer. The distribution of the fluctuations recorded during 2-hour period at midday are also presented in figure 2(b). Between both instruments a very similar distribution of all the fluctuations is obtained with same shape and amplitude for all the parameters considered here. Figure 2(c) presents a comparison of smoothed power spectra between both systems for 2 hours measurements at midday for wind components and potential temperature. The comparison between the TS spectra and the tower spectra is generally quite good and both spectra show the expected -5/3 slope at higher frequencies.

For those fluctuation measurements at 10 Hz, several 2nd order moments can be determined. The following subsection presents the validation of variances of the three components of the wind, of the temperature and of the turbulent sensible heat flux. For all the data, the eddy correlation method is used.

#### 3.1 Variance

10

The variance is commonly used for studying some thermodynamical parameters in the boundary layer because it allows to characterize the dispersion around mean values and can be linked to the intensity of the turbulence. Figures 3(a) and (b) present the comparison of the variance of vertical velocity and temperature calculated every 20 minutes during 10 hours between the fixed sonic anemometer on the mast and the TS. The dashed line represents the difference in altitude between both instruments. Note that the position of the tethered balloon varies from a few meters to tens of meters because of turbulent motions of the atmosphere. That is why the variation of altitude around 60 meters is greater in the middle of the day, when the convection is the strongest. During the afternoon, when the difference in altitude is often greater than 10 meters, the values in  $\sigma_w^2$  is higher for the TS while the values in  $\sigma_t^2$  is lower for the TS. This is consistent with the behaviour of a convective ABL in which the fluctuations of temperature are larger near the surface while fluctuations of vertical wind are more important in the middle of ABL. Regarding the variances of the horizontal components of the wind (not shown here) no trend is observed between the two

10

4