# Peer review of "Turbulence measurements with a tethered balloon"

_Atmospheric Measurement Techniques, 2015_

## Referee Comment (RC1)

Review of:
**Turbulence measurements with a tethered balloon**
Atmospheric Measurement Techniques
Manuscript amt-2015-386
by G. Canut, F. Couvreux, M. Lothon, D. Legain, B. Piguet, A. Lampert, and E. Moulin

**Summary:**

In this manuscript, the authors present measurements from a tethersonde turbulence platform, where a sonic anemometer and inertial motion sensor are deployed on a tethered balloon to calculate turbulent fluxes from near the surface up to $z = 1000$ m in the atmospheric boundary layer. The authors intercompare the tethersonde data with *in situ* tower measurements. While this technique has the potential to augment other in situ and remote sensing techniques to probe the structure of the atmospheric boundary layer, I have a number of concerns about the manuscript. The authors misrepresent what they are doing—the abstract makes it appear that this is the first time turbulence data has been collected from a tethered balloon, but they cite references to make it clear this is not the case. In many places the manuscript is vague (or sometimes incorrect), e.g. the introduction refers to "boundary-layer processes" repeatedly in a vague sense. My largest concern is the validation of the tethersonde platform. The tethersonde was deployed $\sim$200 m horizontally from the tower, but the authors do not attempt to estimate the integral lengthscales to convince the reader that the the turbulence can be expected to remain well correlated at the two measurement locations. They report in Fig. 3 up to 50% differences in fluxes between the balloon and tethersonde measurements. Perhaps the measurement platform can collect perfectly fine data, but I am not convinced of this on the basis of what is presented in this manuscript. I would not consider 50% error to be an acceptable validation. Furthermore, the manuscript does not meet my standards for clear technical English—there are a number of grammatical mistakes, errors in units, misspellings, awkward phrasing, and a general lack of attention to detail. Given the sum total of all these concerns, I cannot recommend this manuscript for publication in *Atmospheric Measurement Techniques*.

**Major comments on article:**

- The abstract does not represent well what the authors are actually doing. The abstract makes the reader think this is the first time turbulence measurements have been collected from a tethered balloon, but the authors cite literature to make it clear this is not the case.

- The authors validate their tethersonde measurements by comparing them with profile tower data. However, this validation is not completely convincing to me. They mentioned that horizontal distances between the two instruments are on the order of 200 m, but never attempted to estimate the integral lengthscales to show that one could expect the turbulence still to be correlated between the two measurement sites. There is some vertical offset as well. They claim good agreement between the tethersonde and the tower data, yet the difference between the fluxes calculated between the two are off by as much as 50%. I do not find this to be good agreement. A sonic anemometer was attached to a balloon together with a GPS inertial motion sensor. The authors say they used the inertial motion sensor to remove the motion of the balloon so their measurements include only a contribution from the turbulence. However,

they fail to provide any details about the motion sensor, and it is unclear to me whether it has the necessary spatial and temporal resolution so that the motion of the balloon platform can be removed to obtain accurate turbulence measurements.

- The English language and grammar in this article need work. There are a number of misspellings, typographical errors, and awkward phrases that do not make sense in English. I have noted some of these passages in my comments below, but ultimately it is the responsibility of the authors (and not of the reviewers) to make sure the paper reads well in clear technical English.

- A number of places in the article lack precision (or are incorrect). In in one place (see comments below) the units employed are incorrect. The authors discuss 'horizontal velocity fluctuations creating TKE,' but this is not an accurate description of how TKE is produced by mean shear—it is the interaction between the vertical momentum flux of streamwise momentum and the mean velocity gradient ($-\overline{uw}\frac{\partial U}{\partial z}$) that produces TKE by mean shear. The introduction refers repeatedly to "turbulent processes" or "boundary-layer processes," in vague terms, but lacks specifics.

**Specific comments**

- **Abstract** "...first deployment of a turbulent probe below a tethered balloon." Is this the first time ever high-resolution turbulence measurements have been taken from a tethersonde platform? If so be clear. The abstract also does not make it clear whether the article is strictly about the deployment and validation of a new instrument platform, or whether scientific questions are being addressed as well.

- **Introduction, first 2 paragraphs** The motivation here is quite vague. "The time evolution of the parameters close to the surface..." (what parameters?) You say "turbulent processes" or "boundary-layer processes" in the introduction several times without being very specific. It would be better to motivate this discussion in terms of transport of momentum, heat, and trace gases ($CO_2$), water vapor, etc. which depends on the stability state of the ABL...

- **1st paragraph, l. 25** necessary is misspelled

- **p. 1, 2nd paragraph** "Understanding turbulent processes in the ABL requires the knowledge of the evolution of the profile of the sensible heat flux." True. But this statement makes it sound like it's the only thing one needs to know about. (What about the momentum flux profile? Velocity variances? Temperature variance? Scalar fluxes?)

- **p. 1, 2nd paragraph** "This platform does not allow [one] to obtain vertical profiles, but only provides some measurements at discrete vertical levels." I don't understand what you mean by this statement. **All measurement techniques** (aircraft data, wind profiling radar, lidar, unmanned aerial vehicles, etc.) have some sort of vertical resolution associated with them, so all of these methods provide discrete measurements. The difference is in the vertical resolution of these measurements.

- **p. 1, 2nd paragraph** "Another *inconvenience* of *an* aircraft platform..."

- **p. 1, 2nd paragraph** "into the heat fluxes characteristics..." This reads awkwardly.

- **p. 2, top paragraph** You cite the Lapworth and Mason (1988) paper, and mention they conducted turbulence measurements with a propeller anemometer from a tethersonde platform, but yet your abstract says that your study presents the "first deployment of a turbulence probe below a tethered balloon in field campaigns." So your abstract is misrepresenting what you are actually doing here. Is this the first study to deploy a tethersonde with a sonic anemometer? If so, make this clear to the reader.

- **p. 2, 2nd paragraph** Now you are describing what you are actually doing, i.e. determining whether a sonic anemometer deployed from a balloon can be used to probe turbulence structure in the ABL. This is what needs to be in the abstract.

- **p. 2, end of 2nd paragraph** "Conclusion ends the paper." Again, not good English.

- **p. 2, Sec. 2** Overview is misspelled.

- **p. 2, Sec. 2.1** "the weight of the cable..." Your units are wrong here.. Weight has units of N (or kg m s$^{-2}$), **not** of kg m$^{-1}$. This is mass per unit length.

- **p. 2, Sec. 2.1** "can be attached to a wide variety of *balloons...*"

- **p. 2, Sec. 2.1** Be careful with capitalization here (names of manufacturers, instrument models...)

- **Sec. 2.1, 2nd paragraph** Does the Gill WindmasterPro provide sonic *virtual* temperature?

- **Sec. 2.1, 2nd paragraph** "A first performance lies in the low weight of the system" This reads awkwardly.

- **p. 3, Sec. 2.2** I find the discussion of the inertial-GPS motion/attitude sensor to be lacking. What is the sampling rate? Spatial resolution? Without this information, I'm unsure whether you're unable to remove the motion of the balloon to obtain accurate turbulence measurements.

- **p. 3, Sec. 3** I know that some horizontal offset between the balloon and tower is unavoidable, but if they are too far apart (i.e. an integral scale or more), the turbulence will not be correlated between the two locations. Did the authors attempt to quantify to consequences of this horizontal offset (e.g. by estimating the integral lengthscales from the autocorrelation function and verifying that they are more than 200 m)? I believe this is something the authors need to address.

- **p. 4, Fig. 2** Do you know why the PDF of $u'$ looks more flat near the peak for the tethersonde than the tower measurements? Also, it appears that the spectra are actually decaying faster than $-5/3$, which I find to be odd.

- **Sec. 3.2** Section heading should be *Fluxes*. There are other places following this in the article where English grammar, punctuation, and readability need to be improved, but this is the responsibility of the authors, not of this reviewer.

- **Sec. 3.2, 1st paragraph** Two things to point out here: 1) eddy covariance is also used to calculate momentum and heat fluxes, not just trace gases. 2) It may be better to denote instantaneous flux as, e.g. $w'a'$, since $x$ is usually used in micromet for streamwise coordinate.

- **Fig. 3** This figure is not very convincing to me as a validation. During some periods there is 50% error (or more!) between the tower and balloon measurements. Perhaps the instrument is working fine and this difference is due to the horizontal and vertical offsets between the tethersonde and tower (or perhaps because of random errors due to the averaging time, see below), but I have a hard time accepting that a 50% error constitutes an acceptable validation. This is my biggest concern with the manuscript.

- **p. 5, 1st paragraph** "The agreement is satisfactory. . . " I do not agree. At times the differences between the measurements is 50% or greater.

- **Sec. 4.1 - TKE profiles** Here you use 20 minute averaging times based on the ogive analysis. This may be enough to reduce systemic bias (e.g. Lenschow et al., *J. Atmos. Ocean. Tech.*, 1994), but I suspect random errors (which are known to increase with height, see chapter by Wynaard in *Workshop on Micrometeorology*, 1973, D. Haugen, Ed.) may be an issue if you're using a 20 minute averaging period throughout the mixed layer. So I'm not entirely convinced that a 20 minute averaging period is long enough.

- **Sec. 5.1** "it is maintained close to 1" What is? Units?

- **Table 2** I usually see articles use $z/z_i$ rather than $z_*$ to denote the ratio between the height and the ABL depth.

- **Figure 7** This figure is not large enough for the axis labels to be legible.

- **Sec. 5.2, 2nd paragraph** "When $A = 0$, the vertical contribution to TKE is zero ($w'^2 = 0$) and the turbulence is only created by horizontal wind fluctuations. This is not a correct statement. In this case, where there is only shear production of TKE and no buoyant production, the shear production term is $-\overline{uw}\frac{\partial U}{\partial z}$, so TKE is being produced by interactions between the momentum flux and the mean wind shear.

- **Secs. 5.2-6** Again, there are a number of places here that read awkwardly and the English needs improvement.

- **Sec. 6** Two comments here. As the authors mention from the literature they cite, this is not the first time a tethersonde has been used for turbulence measurements. They should be careful to accurately represent what they are doing. I do not agree with their point about very good consistency with traditional turbulence measurements. While the system may indeed work correctly, the authors do not manage to convince me of this when they show differences in the fluxes of 50% or more between the *in situ* and tethersonde measurements.

---

## Referee Comment (RC2) · Anonymous Referee #2 · 10 Feb 2016

I found the subject of this paper extremely interesting and timely. Very recently I was discussing observed turbulence profiles in the atmospheric boundary layer with a colleague. As neither of us were aware of how these measurements were originally performed, we speculated on how they might have acquired them with the technology available at the time and discussed how one might be able to make improved measurements with the technology that is currently available. One of the concepts that we discussed was very similar to the system discussed in this manuscript. I am happy to see that such a concept has been developed. The possibilities of this measurement system are very exciting. This system makes it possible to deploy sonic anemometers in relatively fixed locations quickly and inexpensively. I had the same principal concerns as the first referee, even before I had read that review. While I agree with the main points, I believe that the system discussed is a significant improvement in

the ability to directly measure atmospheric turbulence and is well worth publishing. Therefore I recommend that the manuscript should be considered for publication if the authors are willing to perform major revisions. Regarding the technical content of the manuscript: The claims of the originality of mounting a turbulence probe on a tethered balloon in the abstract are obviously too general. The capabilities of the new system needs to be discussed in relation to previous balloon-based turbulence measurement platform discussed in Lapworth and Mason (1988). Their system used three propeller anemometers, therefor the 3D sonic anemometer should easily outperform the Lapworth and Mason's system in more than the total weight, such as frequency response and minimum detectable velocity, etc. The validation of the balloon-mounted system needs a more direct comparison with a tower-mounted sonic anemometer. It should not be necessary to have the turbulence probe mounted to tethered balloon in order to validate the system. It can easily be accomplished by mounting your system to a tower or tripod while still allowing it to pivot and move in the wind. It should be located within a few meters of a fixed 3D sonic anemometer at the same height above the surface. It wouldn't be necessary to mount the two instruments very high above the surface, perhaps just 2 or 3 meters. The two instruments need only measure essentially the same flow, making it obvious whether or not the motion correction is working correctly. While such a comparison would not be exercising the ability of the inertial GPS motion sensor for large scale motions, it is likely that it is more difficult correct for the small-scale motions than it is the large motions due to the movement of the balloon. Editorial concerns: This manuscript needs a thorough copy edit by a native English speaker. There are numerous grammatical and spelling errors throughout the manuscript as well as several instances where words were used incorrectly. For example, the word 'weighted' was used instead of 'weighed' on page 2, left column, line 16 and elsewhere in the manuscript. Page 4, right column, Line 8: You refer to potential temperature as t when $\theta$ is used elsewhere.

---

## Editor Comment (EC1) · E. Pardyjak (Editor) · 26 Apr 2016

I would like to make the author's aware of the following paper (in this journal, AMT) in which 3D sonic measurements below a tethered balloon were reported: Stevens, W. R., Squier, W., Mitchell, W., Gullett, B. K., & Pressley, C. (2013). Measurement of motion corrected wind velocity using an aerostat lofted sonic anemometer. Atmospheric Measurement Techniques Discussions, 6(1), 703–720. http://doi.org/10.5194/amtd-6-703-2013

Please be sure to revise abstract comments and include the reference.

---

## Author Comment (AC1) · 26 Apr 2016

1]CANUT Guylaine 1]COUVREUX Fleur 2]LOTHON Marie 1]LEGAIN Dominique 1]PIGUET Bruno 3]LAMPERT Astrid 1]MOULIN Eric

[1]CNRM-GAME (Météo-France and CNRS), Toulouse, France [2]Laboratoire d'Aerologie, University of Toulouse, CNRS, France [3]Institute of Flight Guidance, TU Braunschweig, Hermann-Blenk-Str. 27, 38108 Braunschweig, Germany

TEXT

TEXT

CANUT Guylaine
(guylaine.canut@meteo.fr)

[Figure]

**Answers to Referee 2:Turbulence measurements with a tethered balloon**

[

April 26, 2016

The authors would like to thank the anonymous reviewer for his/her suggestions and relevant remarks. The text from the review is given in black and the answers in red.

**1 Referee 2**

I found the subject of this paper extremely interesting and timely. Very recently I was discussing observed turbulence profiles in the atmospheric boundary layer with a colleague. As neither of us were aware of how these measurements were originally performed, we speckulated on how they might have acquired them with the technology available at the time and discussed how one might be able to make improved measurements with the technology that is currently available. One of the concepts that we discussed was very similar to the system discussed in this manuscript. I am happy to see that such a concept has been developed. The possibilities of this measurement system are very exciting. This system makes it possible to deploy sonic anemometers in relatively fixed locations quickly and inexpensively. I had the same principal concerns as the first referee, even before I had read that review. While I agree with the main points, I believe that the system discussed is a significant improvement in the ability to directly measure atmospheric turbulence and is well worth publishing.

Therefore I recommend that the manuscript should be considered for publication if the authors are willing to perform major revisions. Regarding the technical content of the manuscript: The claims of the originality of mounting a turbulence probe on a tethered balloon in the abstract are obviously too general. The capabilities of the new system needs to be discussed in relation to previous balloon-based turbulence measurement platform discussed in Lapworth and Mason (1988). Their system used three propeller anemometers, therefor the 3D sonic anemometer should easily outperform the Lapworth and Mason's system in more than the total weight, such as frequency response and minimum detectable velocity, etc. The validation of the balloon-mounted system needs a more direct comparison with a tower-mounted sonic anemometer. It should not be necessary to have the turbulence probe mounted to tethered balloon in order to validate the system. It can easily be accomplished by mounting your system to a tower or tripod while still allowing it to pivot and move in the wind. It should be located within a few meters of a fixed 3D sonic anemometer at the same height above the surface. It wouldn't be necessary to mount the two instruments very high above the surface, perhaps just 2 or 3 meters. The two instruments need only measure essentially the same flow, making it obvious whether or not the motion correction is working correctly. While such a comparison would not be exercising the ability of the inertial GPS motion sensor for large scale motions, it is likely that it is more difficult correct for the small scale motions than it is the large motions due to the movement of the balloon.

Editorial concerns: This manuscript needs a thorough copy edit by a native English speaker. There are numerous grammatical and spelling errors throughout the manuscript as well as several instances where words were used incorrectly. For example, the word "weighted" was used instead of "weighed" on page 2, left column, line 16 and elsewhere in the manuscript. Page 4, right column, Line 8: You refer to potential

temperature as t when $\theta$ is used elsewhere.

**2 Answers**

We appreciate that the referee considers our research topic and sensor as interesting. We would like to thank the referee again for the encouragement to submit an improved revised version. Many answers given to the first referee can improve the manuscript as you suggest. We completely modified the abstract to indicate that this is the first time that a sonic anemometer is embarked above a tethered balloon. The previous attempt of turbulence measurement by a tethered balloon used a propeller anemometer. We propose the following abstract:

*This study presents the first deployment in field campaigns of a turbulence probe built around a sonic anemometer and an inertial motion sensor and suspended below a tethered balloon. This system allows to measure at high frequency the temperature and the horizontal and vertical wind to estimate the turbulent heat fluxes, momentum fluxes as well as turbulent kinetic energy in the lower part of the boundary layer. It has been validated during three campaigns with different convective boundary-layer conditions using turbulent measurements from atmospheric towers and aircrafts.*

Also, we added in the manuscript more details on the accuracy of the GPS-INS and sonic given by the manufacturers. The maximum sampling rate of the inertial-GPS motion is 100hz. Our onboard acquisition system records at 10 Hz both the Inertial-GPS motion/altitude sensor and the sonic anemometer.

As you suggested, we conducted a series of test to assess the capability on this system to remove the motion of the anemometer to compute accurately the wind fluctuations at

(a)

(b)

[Figure]

**Fig. 1.** PSD of TS: (a) the raw anemometer measurement; (b) the computed wind.

a frequency suitable for turbulence studies. In June 2010 the system was suspended below a gantry and left oscillating starting at 30° from the vertical; and we verified that the oscillation was not visible in the computed wind. As a routine monitoring, for each flight we compare the PSD of the raw and corrected wind components. Figure 1 is an example during the test flight in Lannemezan, on August 30 2010 between 14 an 15 h UTC: on the left hand side the raw anemometer measurement, with a clear peak at the modal oscillation frequency of the system (0,2 Hz), on the right hand side the computed wind which exhibits a very linear spectrum. We propose to insert this figure as an annex to a revised manuscript.

We take your Editorial concerns into account. We will improve the paper and a native English speaking professional will revise the manuscript. We corrected "weighed" on page 2, and also we standardized on all the text the notation of the potential temperature to $\theta$.

---

## Author Comment (AC2) · 26 Apr 2016

1]CANUT Guylaine 1]COUVREUX Fleur 2]LOTHON Marie 1]LEGAIN Dominique 1]PIGUET Bruno 3]LAMPERT Astrid 1]MOULIN Eric

[1]CNRM-GAME (Météo-France and CNRS), Toulouse, France [2]Laboratoire d'Aerologie, University of Toulouse, CNRS, France [3]Institute of Flight Guidance, TU Braunschweig, Hermann-Blenk-Str. 27, 38108 Braunschweig, Germany

TEXT

TEXT

CANUT Guylaine
(guylaine.canut@meteo.fr)

11

**Answers to Referee 1:Turbulence measurements with a tethered balloon**

[

April 26, 2016

The authors would like to thank the anonymous reviewer for his/her suggestions and relevant remarks. The text from the review is given in black and the answers in red.

**1  Major comments on article**

The abstract does not represent well what the authors are actually doing. The abstract makes the reader think this is the first time turbulence measurements have been collected from a tethered balloon, but the authors cite literature to make it clear this is not the case.

You are right indeed. We completely modified the abstract to indicate that this is the first time that a sonic anemometer is embarked above a tethered balloon. The previous attempt of turbulence measurement by a tethered balloon used a propeler anemometer. We propose the following abstract as :
*This study presents the first deployment in field campaigns of a turbulence probe built*

*around a sonic anemometer and an inertial motion sensor and suspended below a tethered balloon. This system allows to measure at high frequency the temperature and the horizontal and vertical wind to estimate the turbulent heat fluxes, momentum fluxes as well as turbulent kinetic energy in the lower part of the boundary layer. It has been validated during three campaigns with different convective boundary-layer conditions using turbulent measurements from atmospheric towers and aircrafts.*

The authors validate their tethersonde measurements by comparing them with profile tower data. However, this validation is not completely convincing to me. They mentioned that horizontal distances between the two instruments are on the order of 200 m, but never attemped to estimate the integral lengthscales to show that one could expect the turbulence still to be correlated between the two measurement sites. There is some vertical offset as well. They claim good agreement between the tethersonde and the tower data, yet the difference between the fluxes calculated between the two are off by as much as 50%. I do not find this to be good agreement. A sonic anemometer was attached to a balloon together with a GPS inertial motion sensor. The authors say they used the inertial motion sensor to remove the motion of the balloon so their measurements include only a contribution from the turbulence. However, they fail to provide any details about the motion sensor, and it is unclear to me whether it has the necessary spatial and temporal resolution so that the motion of the balloon platform can be removed to obtain accurate turbulence measurements.

We agree on the use of the integral length scale as a characteristics of the turbulence. See below our complete answer to your comment on relative to p.3 Sec.3.
It was our deliberate choice to describe the retrieved atmospheric quantities rather than the instrumental set-up, but we understand the need to be convincing on this issue. Our modified text is presented later, in the answer to the comment on p. 3, Sec. 2.2.

The English language and grammar in this article need work. There are a number of

misspellings, typographical errors, and awkward phrases that do not make sense in English. I have noted some of these passages in my comments below, but ultimately it is the responsibility of the authors (and not of the reviewers) to make sure the paper reads well in clear technical English.

We have taken notes of this remark. The last version of the manuscript will be revised by a native English speaking professional.

A number of places in the article lack precision (or are incorrect). In in one place (see comments below) the units employed are incorrect. The authors discuss "horizontal velocity fluctuations creating TKE," but this is not an accurate description of how TKE is produced by mean shear—it is the interaction between the vertical momentum flux of streamwise momentum and the mean velocity gradient ($-uw\delta U\ \delta Z$) that produces TKE by mean shear. The introduction refers repeatedly to "turbulent processes" or "boundary-layer processes," invague terms, but lacks specifics.

We changed the TKE discussion as suggested. See below our complete answer to your comments to p5. Sec.5.2.

**2   Specific comments**

**Abstract** ". . . first deployment of a turbulent probe below a tethered balloon." Is this the first time ever high-resolution turbulence measurements have been taken from a tethersonde platform? If so be clear. The abstract also does not make it clear whether the article is strictly about the deployment and validation of a new instrument platform, or whether scientific questions are being addressed as well.

Indeed, this is not the first time turbulence measurements have been made with a

tethersonde platform. However, the previous measurements realised by Lapworth and Mason (1988) used a propeller anemometer to acquire the three wind component without the temperature. This is indeed the first time that a sonic anemometer has been mounted with an inertial motion sensor on a tethersonde. See the proposition of a new abstract at the page 1.

**Introduction**, first 2 paragraphs The motivation here is quite vague. "The time evolution of the parameters close to the surface. . . " (what parameters?) You say "turbulent processes" or "boundary-layer processes" in the introduction several times without being very specific. It would be better to motivate this discussion in terms of transport of momentum, heat, and trace gases ($CO_2$), water vapour, etc. which depends on the stability state of the ABL.

We propose to modify the first paragraph of the introduction as follows :
*Rare observations allow to sample vertically the turbulent fluxes in the boundary layer. These observations are however crucial as their divergence controls the time evolution of mean parameters such as temperature, water vapour and wind. A large number of observational studies of heat and momentum fluxes in the ABL have been carried out in recent year. These have utilized three types of support in order to place the measuring instruments in the airflow above the surface. The two main platforms used are the instrumented mast and the research aircraft.*

**1st paragraph, l. 25** necessary is misspelled
This has been corrected

**p. 1, 2nd paragraph** "Understanding turbulent processes in the ABL requires the knowledge of the evolution of the profile of the sensible heat flux." True. But this statement makes it sound like it's the only thing one needs to know about. (What about the momentum flux profile? Velocity variances? Temperature variance? Scalar

fluxes?)

Indeed, this is not the only parameter to know in the ABL to understand the turbulent processes in the ABL. We modified this sentence as:

*The turbulence in the ABL and its impact on mean thermodynamical variables such as temperature, water vapour mixing ratio and winds can be estimated via the turbulent fluxes. In this paper we focus on sensible heat fluxes, momentum fluxes as well as turbulent kinetic energy.*

**p. 1, 2nd paragraph** "This platform does not allow [one] to obtain vertical profiles, but only provides some measurements at discrete vertical levels." I don't understand what you mean by this statement. All measurement techniques (aircraft data, wind profiling radar, lidar, unmanned aerial vehicles, etc.) have some sort of vertical resolution associated with them, so all of these methods provide discrete measurements. The difference is in the vertical resolution of these measurements.

Indeed tethersonde also provides some measurements at discrete vertical levels, however compared to aircraft data, it has the advantages to i/ be much cheaper to implement, ii/ provide local informations (in order to derive fluxes aircraft measurements have to be obtained over flight legs of at least 20-40 km). We replaced in the revised manuscript as:

*Previous studies (Lenschow and Stankov (1986), Saïd et al. (2010)) used instrumented aircraft to measure turbulent heat flux in altitude. Aircraft are costly, are often constrained by a minimal flight altitude, and their relative air speed imposes to use fast-response instruments in order to measure variations at physical scales equivalent to the ones measured from a fixed location.*

**p. 1, 2nd paragraph** "Another inconvenience of an aircraft platform. . . " We replaced inconvenient by inconvenience

**p. 1, 2nd paragraph** "into the heat fluxes characteristics. . . " This reads awkwardly. We propose to change all the sentence with : *Fixed-point tower measurements have largely been used to provide useful characterization of heat fluxes above the surface layer. However, these measurements are limited in height with only a few towers reaching more than 100 m. The tethered balloon allows the sampling of the atmosphere up to 800m.*

**p. 2, top paragraph** You cite the Lapworth and Mason (1988) paper, and mention they conducted turbulence measurements with a propeller anemometer from a tethersonde platform, but yet your abstract says that your study presents the "first deployment of a turbulence probe below a tethered balloon in field campaigns." So your abstract is misrepresenting what you are actually doing here. Is this the first study to deploy a tethersonde with a sonic anemometer? If so, make this clear to the reader. Indeed, we modified the abstract in order to state that the originality of this study is the use of sonic anemometer mounted on a tethered balloon.

**p. 2, 2nd paragraph** Now you are describing what you are actually doing, i.e. determining whether a sonic anemometer deployed from a balloon can be used to probe turbulence structure in the ABL. This is what needs to be in the abstract. We changed the abstract to add the information that this study is the first one to use a sonic anemometer mounted on a tethered sonde to sample turbulence. Also, we added the motivations for the development of such probe.

**p. 2, end of 2nd paragraph** "Conclusion ends the paper." Again, not good English. We changed with : *Conclusions are given in section 6.*

**p. 2, Sec. 2** Overview is misspelled. We corrected Overview

**p. 2, Sec. 2.1** "the weight of the cable. . . " Your units are wrong here.. Weight has units of N (or $kgms^{-2}$), not of $kgm^{-1}$. This is mass per unit length. We agree with your comment. We replace weight by mass.

**p. 2, Sec. 2.1** "can be attached to a wide variety of balloons. . . " We added 's' at the end of balloon.

**p. 2, Sec. 2.1** Be careful with capitalization here (names of manufacturers, instrument models. . . ) We corrected Mti-G from Xsens by MTi-G from Xsens

**Sec. 2.1, 2nd paragraph** Does the Gill WindmasterPro provide sonic virtual temperature?
The Gill WindmasterPro does not provide directly the virtual temperature. The virtual temperature defined by meteorologists is the temperature at which dry air has the same density as moist air at the same pressure. Kaimal J.C. and Gaynor J. E. (1990, Another look at sonic thermometry) compared the sonic temperature measured by sonic anemometer with the virtual temperature and concluded that in applications where the virtual temperature is needed, to include the buoyant contribution from water vapor, the sonic temperature could be used with negligible loss of accuracy without concurrent measurement of humidity fluctuations to convert temperatures to virtual temperatures. Therefore the sonic temperature can be used as a good proxy of the virtual temperature and this is what we have been doing in this study.

**Sec. 2.1, 2nd paragraph** "A first performance lies in the low weight of the system" This reads awkwardly. We propose to rephrase this sentence :
*Major advantage of the instrumentation proposed here is its relatively low weight, 2kg against 10kg for the Lapworth platform (Lapworth 1988).*

**p. 3, Sec. 2.2** I find the discussion of the inertial-GPS motion/attitude sensor to be lacking. What is the sampling rate? Spatial resolution? Without this information, I'm unsure whether you're unable to remove the motion of the balloon to obtain accurate turbulence measurements.

We agree with your remarks. We added this information to the paper. We also added a table to give the accuracy of the GPS-INS motion sensor and sonic anemometer given by the manufacturers. The maximum sampling rate of the inertial-GPS motion sensor is 100hz. Our onboard acquisition system records at 10 Hz both instrument. A series of test were conducted to assess the capability on this system to remove the motion of the anemometer to compute accurately the wind fluctuations at a frequency suitable for turbulence studies. In june 2010 the system was suspended below a gantry and left oscillating starting at 30° from the vertical; and we verified that the oscillation was not visible in the computed wind. As a routine monitoring, for each flight we compare the PSD of the raw and corrected wind components. Below is an example during the test flight in Lannemezan, on August 30 2010 between 14 an 15 h UTC: on the left hand side the raw anemometer measurement, with a clear peak at the modal oscillation frequency of the system (0,2 Hz), on the right hand side the computed wind which exhibits a very linear spectrum. We propose to insert this figure as annex to a revised manuscript.

**p. 3, Sec. 3** I know that some horizontal offset between the balloon and tower is unavoidable, but if they are too far apart (i.e. an integral scale or more), the turbulence will not be correlated between the two locations. Did the authors attempt to quantify to consequences of this horizontal offset (e.g. by estimating the integral lengthscales from the autocorrelation function and verifying that they are more than 200 m)? I believe this is something the authors need to address.

We calculated the integral lengthscale of w, we obtain 81m and 84m respectively for the tower and the tethered balloon. This is shown in Table 1 that also indicates integral lengthscale for other variables. The values are very similar in-between the tower and the tethered balloon indicating a similar turbulence sampled by both instruments. In our opinion, the similarity of the value is more important that the specific values which depend on the length of the sample, as show by Durand et al. (2000)(Turbulent length-scales in the marine atmospheric mixed layer, Quarterly Journal of the Royal Meteorological Society 2000).

|          | w  | u   | v   | t   |
|----------|-----|-----|-----|-----|
| $L_{int}$ TS  | 84 | 510 | 300 | 321 |
| $L_{int}$ mat | 81 | 552 | 360 | 311 |

Table 1: Integral length scale of w, u, v and t in Lannemezan, on August 31 2010 between 12 an 14h UTC

**p 4, Fig 2** Do you know why the PDF of u looks more flat near the peak for the tethersonde than the tower measurements? Also, it appears that the spectra are actually decaying faster than

$$-$$

$5/3, which I find to be odd.$

Concerning the PDF of u, we have no explanation why the PDF is flatter near the peak for the TS than the tower measurements. To try to document the phenomena, we ploted the PDF for other time and other campaign on figure 2. Below is two different examples. One during the same day that this presented on the manuscript (between 16 and 17h) and a second during the campaign Bourges 2013. These exemples show that the behaviour of the PDF of u on the exemple presented on the manuscript is not systematic.

We would like to thank the referee for the remark about the decaying faster than -5/3. We made a mistake when we plotted the line in -5/3. The dashed line do not

corresponding at the -5/3 shape. In a revised version of the manuscript the shape will be corrected (see Figure 3).

**Sec. 3.2 Section heading should be Fluxes.** There are other places following this in the article where English grammar, punctuation, and readibility need to be improved, but this is the responsibility of the authors, not of this reviewer. We take your remarks into accounts. We will improve the paper and a native English speaking professional will revise the manuscript.

**Sec. 3.2,** 1st paragraph Two things to point out here: 1) eddy covariance is also used to calculate momentum and heat fluxes, not just trace gases. 2) It may be better to denote instantaneous flux as, e.g. w a , since x is usually used in micromet for streamwise coordinate. We agree with you. We changed the sentence as : *Eddy covariance is a well established method for the direct measurement of the vertical exchange of heat, mass and momentum in the atmosphere.* We also changed 'x' by 'a' when we describe the flux.

**Fig. 3** This figure is not very convincing to me as a validation. During some periods there is 50% error (or more!) between the tower and balloon measurements. Perhaps the instrument is working fine and this difference is due to the horizontal and vertical offsets between the tethersonde and tower (or perhaps because of random errors due to the averaging time, see below), but I have a hard time accepting that a 50% error constitutes an acceptable validation. This is my biggest concern with the manuscript. Most differences occurred when the differences in altitude between the tower and the TS are the more important (between 10 and 20 meters). It's the reason why we added this information on the figure. Nevertheless, the correlation coefficient between both dataset is always larger than 0.8. This mean a correct correlation between the tower measurement and the TS.

**p. 5, 1st paragraph** "The agreement is satisfactory. . . " I do not agree. At times the differences between the measurements is 50% or greater. We propose to modify this sentence and to remove the agreement is satisfactory.

**Sec. 4.1** - TKE profiles Here you use 20 minute averaging times based on the ogive analysis. This may be enough to reduce systemic bias (e.g. Lenschow et al., J. Atmos. Ocean. Tech., 1994), but I suspect random errors (which are known to increase with height, see chapter by Wynaard in Workshop on Micrometeorology, 1973, D. Haugen, Ed.) may be an issue if you're using a 20 minute averaging period throughout the mixed layer. So I'm not entirely convinced that a 20 minute averaging period is long enough.

In recent book ( Aubinet et al., Eddy covariance, A practical guide to measurement and data Analysis, ed. Springer,2012) we found that the averaging period is satisfactory if the value of the integral approaches a constant value at low frequencies. With this method we found a constant value around 16 minutes. In addition to the ogive analysis, we performed several tests on the averaging times. The Figure 4 above shows for one day of the BLLAST campaign the values of TKE for several averaging time periods (5, 10, 20 and 40 minutes). We show a large variability of the value for the period smaller than 20 minutes. Instead, we see few variability of the value for the 40 minutes averaging period.

**Sec. 5.1** "it is maintained close to 1" What is? Units? We forgot the unit. We added $m^2s^{-2}$ in the text.

**Table 2** I usually see articles use $z/zi$ rather than $z$ to denote the ratio between the height and the ABL depth. We think that both notation are used to denote the ratio between the height and the ABL depth. But we change $z_*$ into $z/zi$.

**Figure 7** This figure is not large enough for the axis labels to be legible. We understand the remarks. We changed the size of the figure.

**Sec. 5.2**, 2nd paragraph "When $A = 0$, the vertical contribution to TKE is zero ($w^2 = 0$) and the turbulence is only created by horizontal wind fluctuations". This is not a correct statement. In this case, where there is only shear production of TKE and no buoyant production, the shear production term is $-uw\delta U/\delta z$ , so TKE is being produced by interactions between the momentum flux and the mean wind shear.

Indeed, we changed the TKE discussion as: *When $A = 0$, the vertical contribution to TKE is zero ($w^2 = 0$) and the horizontal wind fluctuations are the main contributor to the TKE.*

**Secs. 5.2-6** Again, there are a number of places here that read awkwardly and the English needs improvement. We take your remarks into account. We will improve the paper.

**Sec. 6** Two comments here. As the authors mention from the literature they cite, this is not the first time a tethersonde has been used for turbulence measurements. They should be careful to accurately represent what they are doing. I do not agree with their point about very good consistency with traditional turbulence measurements. While the system may indeed work correctly, the authors do not manage to convince me of this when they show differences in the fluxes of 50% or more between the in situ and tethersonde measurements.

Like we proposed to modify the abstract, we changed the conclusion in the revised manuscript to clarify that it is well the first time that a sonic anemometer is suspended below a tethered balloon. Concerning the agreement between traditional turbulence measurement and this new system, we think that the higher correlation coefficients

obtained in this study show the capability of this innovative sensor.

(a) (b)

[Figure]

**Fig. 1.** PSD of TS: (a) the raw anemometer measurement; (b) the computed wind.

[Figure]

(a)          (b)

pdf_2010_test.png     pdf_2013_test.png

**Fig. 2.** Fluctuation distribution of w, v, u and t of (a) 1-hour sample on 31 agust 2010 and (b) 50 minutes sample on 28 August 2013.

[Figure]

/home/canut/figure/papier/spectre_pente_corrige_201

**Fig. 3.** Power spectra density correspond to the 10 hz time series of a 2-hour sample on 31 august 2010.

[Figure]

**Fig. 4.** Estimates of TKE for one day of the BLLAST campaign with several time averaging periods.

---

## Author Comment (AC5) · 27 Apr 2016

Dear Editor, We become aware of this article, in which the authors present a sensor with large technical similarity to ours, but don't investigate its capacity for turbulence measurements. However, this article has been in discussions mode without authors' answers to comments for 3 years. The status on AMTD website is 'peer review stopped'. We agree to add in the revised manuscript some information like: "In recent years, a team of the US Environmental Protection Agency office of research and development developed a similar system to estimate the windspeed and direction in altitude." Nevertheless, We do not regard as opportune to include a reference to a paper which did not complete the peer-review process.